# Assessment of Satellite AOD during the 2020 Wildfire Season in the Western U.S.

**Xinxin Ye** [1],*[iD], **Mina Deshler** [1],†, **Alexi Lyapustin** [2][iD], **Yujie Wang** [3], **Shobha Kondragunta** [4] **and Pablo Saide** [1,5]

1   Department of Atmospheric and Oceanic Sciences, University of California, Los Angeles, CA 90095, USA
2   Laboratory for Atmospheres, NASA Goddard Space Flight Center, Greenbelt, MD 20771, USA
3   Joint Center for Earth Systems Technology, University of Maryland Baltimore County,
    Baltimore, MD 21228, USA
4   NOAA NESDIS Center for Satellite Applications and Research, College Park, MD 20740, USA
5   Institute of the Environment and Sustainability, University of California, Los Angeles, CA 90095, USA
\*   Correspondence: xinxinye@g.ucla.edu
†   Current address: Los Alamos National Laboratory, Los Alamos, NM 87545, USA.

**Abstract:** Satellite remote sensing of aerosol optical depth (AOD) is essential for detection, characterization, and forecasting of wildfire smoke. In this work, we evaluate the AOD (550 nm) retrievals during the extreme wildfire events over the western U.S. in September 2020. Three products are analyzed, including the Moderate-resolution Imaging Spectroradiometers (MODIS) Multi-Angle Implementation of Atmospheric Correction (MAIAC) product collections C6.0 and C6.1, and the NOAA-20 Visible Infrared Imaging Radiometer (VIIRS) AOD from the NOAA Enterprise Processing System (EPS) algorithm. Compared with the Aerosol Robotic Network (AERONET) data, all three products show strong linear correlations with MAIAC C6.1 and VIIRS presenting overall low bias (<0.06). The accuracy of MAIAC C6.1 is found to be substantially improved with respect to MAIAC C6.0 that drastically underestimated AOD over thick smoke, which validates the effectiveness of updates made in MAIAC C6.1 in terms of an improved representation of smoke aerosol optical properties. VIIRS AOD exhibits comparable uncertainty with MAIAC C6.1 with a slight tendency of increased positive bias over the AERONET AOD range of 0.5–3.0. Averaging coincident retrievals from MAIAC C6.1 and VIIRS provides a lower root mean square error and higher correlation than for the individual products, motivating the benefit of blending these datasets. MAIAC C6.1 and VIIRS are further compared to provide insights on their retrieval strategy. When gridded at 0.1° resolution, MAIAC C6.1 and VIIRS provide similar monthly AOD distribution patterns and the latter exhibits a slightly higher domain average. On daily scale, over thick plumes near fire sources, MAIAC C6.1 reports more valid retrievals where VIIRS tends to have retrievals designated as low or medium quality, which tends to be due to internal quality checks. Over transported smoke near scattered clouds, VIIRS provides better retrieval coverage than MAIAC C6.1 owing to its higher spatial resolution, pixel-level processing, and less strict cloud masking. These results can be used as a guide for applications of satellite AOD retrievals during wildfire events and provide insights on future improvement of retrieval algorithms under heavy smoke conditions.

**Keywords:** aerosol optical depth; MODIS; VIIRS; retrieval; wildfire smoke

---

## 1. Introduction

    Space-based remote sensing of atmospheric aerosol optical depth (AOD) is an essential means to characterize the spatial and temporal distributions of aerosols in the perspective of total column loading continuously and globally. Over the western U.S., large amounts of biomass burning aerosols are emitted from the increasingly severe and frequent wildfire events [1]. Wildfire smoke, which mainly comprises organic carbon (OC), black carbon (BC) and gaseous species that contribute to the formation of secondary species such as secondary organic aerosols (SOA) and ozone ($O_3$), has proven to have significant adverse

---

impacts on ecology, air quality and human health [2]. For hazardous wildfire events, AOD remote sensing products have been extensively applied to support the detection and monitoring of fire intensity, air pollution, and visibility, inform numerical weather forecasting through data assimilation and model evaluation [3–5], and deliver valuable information for environmental decision makers. In addition, satellite AOD data have been leveraged in estimating the fine particulate matter ($PM_{2.5}$) concentrations near the land surface to evaluate and predict the associated health exposures [6–8], as well as in providing objective top-down constraints for the quantification of smoke emissions [9–11]. Therefore, the accuracy and precision of AOD retrievals are crucial for studies on fire and smoke.

Satellite retrievals of AOD and other aerosol properties have been realized by using reflected solar radiance measurements in visible and near infrared bands since the 1970s. Radiance measurements made from the twin Moderate-resolution Imaging Spectroradiometers (MODIS), onboard the Earth Observing System (EOS) Terra and Aqua platforms, launched in December 1999 and May 2002, respectively, provide global AOD records spanning over two decades. As a successor to MODIS, the Visible Infrared Imaging Radiometer Suite (VIIRS) is a moderate-resolution imaging radiometer, onboard polar-orbiting Suomi National Polar-orbiting Partnership (SNPP, October 2011~) and the National Oceanic and Atmospheric Administration-20 (NOAA-20, November 2017~) satellites with future launches planned. It passively detects reflectance at multiple visible and infrared wavelengths and provides AOD products at a nadir resolution of 750 m [12,13], which is higher compared to MODIS AOD products [14]. In addition, the wider across-track swath of VIIRS (3060 km vs. 2330 km for MODIS) allows for full global daily coverage, while gaps are observed with MODIS in the tropics. With three scheduled follow-ons to NOAA-20, VIIRS offers the opportunity to extend the geophysical data records of EOS MODIS into the next decades [15].

Several AOD retrieval algorithms and products have been developed and improved over the last decades. The Dark Target (DT) [16–18] and the Deep Blue (DB) [19,20] algorithms are extensively used for different sensors and widely evaluated. More recently, the Multi-Angle Implementation of Atmospheric Correction (MAIAC) algorithm [21,22] was developed for MODIS Terra and Aqua data which increased spatial resolution of AOD retrievals from ~10 to 1 km. The higher spatial resolution of MAIAC product allows better data coverage than coarser products such as MODIS DB (10 km) and DT (10 km, 3 km) [23]. Different algorithm products are validated and compared in regional [23–27] and global studies [28–30]. With respect to the AErosol RObotic NETwork (AERONET) data, the performance of MODIS DT, DB, and MAIAC retrievals are comparable over dark surfaces over eastern North America (NA) [23]. While over the western NA, with the steep changes in topography and abundance of bright surfaces, MODIS DB and MAIAC algorithms both show lower biases than DT, with MAIAC offering the smallest spread of errors [23]. The DT products are found to exhibit systematic positive biases at higher surface reflectance, as seen for both the MODIS DT retrievals [23] and the SNPP VIIRS operational DT algorithm by NOAA [12,25]. This is mainly because that the DT algorithm was developed for the retrieval over vegetated surfaces with the assumption that aerosols brighten the measured signal, which is violated over bright surface. A more recent evaluation suggests that MAIAC presents better statistical metrics than MODIS DB and VIIRS DB products by NASA over the western U.S. [24]. Overall, MAIAC shows low bias over a wide range of surface conditions and view geometry [25], and a higher consistency between Terra and Aqua MODIS AODs than those from DB and DT algorithms, owing to the algorithm differences and MAIAC's enhanced calibration of MODIS data [30,31].

Despite the significance and advances, deficiencies of satellite products have been noticed over optically thick plumes, which may present high visible reflectance and spatial variability, causing unintended exclusion of high-AOD over heterogenous plumes [24,32,33]. For example, during the 2015 Indonesian fire event [34], the operational DT algorithm is reported to be biased low by about 0.22 for regional AOD, and up to 3.0 for certain 0.5° grid boxes [32]. Additionally, the variety in smoke optical properties could lead to bias in retrievals [35,36]. The errors and

gaps in the AOD retrievals can limit smoke emission estimation from satellites [11]. These imply that specific improvements are necessary. Attempts have been made by using a specific aerosol model using the AERONET data, cloud masking based on the MODIS cloud optical properties algorithm, and relaxed thresholds on both inland water tests and upper limits of the AOD retrieval, resulting in the improved agreement of retrieved AODs with AERONET [32]. Due to the recently reported underestimation of AOD over the most intense smoke plumes [24], updates regarding smoke aerosol model are implemented into the MAIAC Collection C6.1 algorithm, but the performance under high aerosol loading is yet to be examined. Specifically, the new MAIAC C6.1 algorithm employs updated regional aerosol models, which were revised based on climatology analysis of AERONET v3 inversion data [37,38] in particular at high AOD. The MAIAC C6.1 re-processing of the MODIS record is currently ongoing and is expected to be released in late 2022. The details of MAIAC C6.1 updates will be published in another paper. This work employed both officially available MAIAC C6.0 data and C6.1 testing data processed using NASA Center for Climate Simulations (NCCS) and provided by the MAIAC team.

In late summer 2020, intensive wildfires occurred along the west coast of the U.S. and resulted in an extremely increased health burden due to smoke aerosols [39,40]. In this paper, we evaluate and compare AOD retrievals from three datasets, i.e., MODIS MAIAC (MCD19A2) C6.0 and C6.1, and NOAA-20 VIIRS AOD (v2r3) derived using the NOAA Enterprise Processing System (EPS) algorithm, during September 2020 over a domain (24 to 52°N, −130 to −100°W) covering the western U.S. We focus on the accuracy and challenges of remote sensing of AOD during heavy wildfire smoke episodes. Following a brief description of the algorithms and datasets used here (Section 2), we first present the validation by direct comparison against the ground-based measurements from 41 AERONET sites in the western U.S., for individual products and collocated records (Section 3.1). As NOAA-20 VIIRS and Aqua MODIS both fly in afternoon sun-synchronized orbits with close equator crossing times and similar sensor characterizations, their AOD products are eligible for a comparison. Thus, we further compared the differences of 0.1° gridded AOD maps from NOAA-20 VIIRS and Aqua MODIS MAIAC C6.1 (Section 3.2) to gain more insights in their overall retrieval differences. In addition, the Cloud-Aerosol Lidar with Orthogonal Polarization (CALIOP) data is used to evaluate cloud masking performance of the AOD products and its possible influence on AOD accuracy. Typical cases are analyzed in Section 3.3 to elaborate the retrievability and differences between Aqua MODIS MAIAC C6.1 and VIIRS AODs. Concluding remarks are given in Section 4.

## 2. Data and Method

### 2.1. Satellite AOD Retrievals

In this section, we present an overview of the characteristics and differences of the satellite AOD products evaluated in this work, which are summarized in Table 1.

**Table 1.** Summary of the AOD products evaluated in this work.

| Dataset | MODIS MAIAC (MCD19A2) C6.0 | MODIS MAIAC (MCD19A2) C6.1 | NOAA-20 VIIRS v2r3 |
|---|---|---|---|
| Equatorial crossing time | 10:30 LST (Terra), 13:30 LST (Aqua) | | 13:30 LST |
| Nadir resolution (km) | 1 km | | 0.75 km |
| Algorithm | MAIAC | | NOAA Enterprise Processing System (EPS) |
| Swath Width (km) | 2330 | | 3060 |
| Surface reflectance | Retrieved using consecutive MODIS overpasses | | Land: assumed spectral relationships and reflectance database; Ocean: surface reflectance model |
| Aerosol Models | 8 regional background models and a dust model | Same as C6.0 with updates for smoke aerosol | Generic, dust, smoke, and urban aerosols |
| Upper limit of AOD | 4.0 (at 470 nm) | 6.0 (at 470 nm) | 5.0 (at 550 nm) |
| Reference | [21,41] | | [13] |

### 2.1.1. MODIS MAIAC C6.0 AOD

In this work, the MODIS Terra and Aqua combined Level 2 aerosol product based on the MAIAC algorithm (MCD19A2 C6.0) [21,41] is used (Table 1), which provides AOD (470 and 550 nm) at 1 km-resolution grid over both dark vegetated surfaces and bright surfaces [22]. Terra and Aqua fly along sun-synchronous orbits at ~705 km altitude with the daytime equatorial crossing times of ~10:30 Local Solar Time (LST) and ~13:30 LST, respectively.

The MAIAC algorithm is an advanced method using time series analysis and a combination of pixel- and image-based processing to improve accuracy of cloud detection, aerosol retrievals, and atmospheric correction [22,42]. Time series analysis of MODIS data is used to retrieve bidirectional reflectance factor (BRF) and spectral regression coefficient (SRC) over both dark vegetated surfaces and bright surfaces [22]. The L1B spectral reflectance data are first calibrated using the standard C6 calibration [43] augmented with polarization correction for Terra MODIS [44], residual detrending, and MODIS Terra-to-Aqua cross calibration [31]. Then, the 4 (at pole) to 16 (at equator) days of clear MODIS L1B measurements are gridded at a fixed 1 km sinusoidal grid and split into 1200 × 1200 km tiles and 25 × 25 km blocks [21]. The consecutive MODIS overpasses allow for collecting observations of a location from multiple angles. Given the daily rate of MODIS observations, the surface reflectance changes slowly in time compared to that of aerosols and clouds. This allows MAIAC to characterize the surface spectral reflectance at 1 km that are required for aerosol retrievals, using the minimum reflectance method. Additionally, accumulation of multi-angle MODIS data from the consecutive overpasses in a sliding time window of 4–16 days allows MAIAC to characterize the surface bidirectional reflectance distribution function (BRDF) at 1 km [21].

For the retrieval of AOD, MAIAC considers eight regional background aerosol models and a dust aerosol model. The model parameters are detailed in [21] and are generally representative of the AERONET regional climatology, with empirical adjustments aimed at achieving a better match of retrieved AOD to AERONET data. A smoke test is used to identify biomass burning aerosols from clouds [42], which relies on the relative increase in aerosol absorption at 412 nm compared to 470–670 nm due to multiple scattering and enhanced absorption by organic aerosols related to fires. The upper limit of AOD at 470 nm in the look-up table (LUT) is 4.0. The accuracy for MAIAC C6.0 AOD is reported globally with 66% (or ±1σ) of retrievals agreeing with AERONET within error envelope (EE) of ±(0.05 + 10% AOD) (AOD refers to AERONET AOD, same hereinafter) [21], better than the standard accuracy of ±(0.05 + 15% AOD) determined for the global DT algorithm over land [18]. An assessment in western NA indicates the bias of 0.015 and RMSE of 0.062 [23]. Over smoke plumes, compared to other algorithms, MAIAC provides more available AOD data due to its capability to distinguish thick smoke, which is frequently identified as clouds by other methods [21]. The quality assessment (QA) in MAIAC data contains cloud mask, adjacency mask, aerosol model, and overall QA flag. A more detailed description of the MAIAC algorithm is given in [21].

### 2.1.2. MODIS MAIAC C6.1 AOD

MODIS MAIAC C6.0 data has proven to show underestimations of AOD for high loading of biomass burning aerosols (AOD > 0.6) [45]. Thus, in the recently developed C6.1 product, specific updates are implemented with focus on improving the retrieval performance for smoke conditions. Updated aerosol models are used based on AERONET climatology analysis, which account for the increased absorption and decreased effective particle size under high AOD conditions for smoke events, effectively removing the bias and further improving the retrieval accuracy. More absorbing smoke aerosol is assumed in C6.1 with the absorption dynamically increasing, and the coarse mode fraction decreases at AOD > 0.6 to mimic particle properties for biomass burning smoke aerosols. These changes generally lead to higher retrieved AODs in C6.1 compared to C6.0. In addition, the upper limit of AOD at 470 nm for C6.1 product is elevated to 6.0 (compared to 4.0 in C6.0), which

allows the algorithm to capture heavy aerosol situations during smoke events. More details about the updates for the MAIAC C6.1 will be provided in a future paper (Lyapustin and Wang, in preparation).

### 2.1.3. NOAA-20 VIIRS AOD

The NOAA's Joint Polar Satellite System (JPSS) Granule (JPSS_GRAN) Level 2 Enterprise Aerosol Optical Depth/Aerosol Particle Size product [46] (v2r3), generated from data of the Visible Infrared Imaging Radiometer Suite (VIIRS) onboard the NOAA-20 satellite, is used in this work. NOAA-20 (previously known as JPSS-1) is the second spacecraft within NOAA's next generation of polar-orbiting satellites, having similar a payload as SNPP. NOAA-20 crosses the equator about 13:30 LST from the sun-synchronous orbit at ~834 km altitude. We note that the SNPP VIIRS EPS algorithm AOD product is also available. The NOAA-20 and SNPP VIIRS AODs are developed using the same instrument but on-board distinct platforms, and it has been reported that the AOD retrieval difference from them is largely contributed by uncertainties in upstream radiometric calibration differences [29]. In this work, we would like to focus on retrieval accuracies under wildfire smoke conditions, for which the representation of surface reflectance and aerosol models used in the algorithms are expected to play a more important role [29]. Thus, we only include NOAA-20 VIIRS data.

The AOD data are retrieved at 550 nm globally in daylight except under cloudy or other unfavorable conditions and reported at multiple wavelengths between 0.4 to 2.25 μm [13]. The NOAA EPS algorithm is applied at pixel level with resolution of 750 m in nadir, by comparing the satellite-observed top-of-atmosphere (TOA) reflectance with those pre-calculated for a set of aerosol models, AODs, and viewing geometries stored in LUTs. Separate approaches are used over ocean and land due to the distinct surface properties and aerosol types. Over land, AOD is retrieved for dark, vegetated surface following the DT algorithm [16,17,47,48]. The surface is assumed to be Lambertian, and the retrieval of surface reflectance is performed simultaneously with AOD using the visible and near infrared channels. Four aerosol models representing generic, dust, smoke, and urban aerosols [13,17] are considered. Over bright and snow-free surfaces, shorter wavelength channels are chosen to retrieve AOD, due to the relatively low interference of surface reflectance in these channels. A static 0.1° by 0.1° spectral surface reflectance ratio dataset is used to characterize the surface contribution [49]. Over ocean, the algorithm is based on [16,49], assuming water surface reflectance being modeled with sufficient accuracy. Four fine-mode and five coarse-mode models have been adopted from MODIS [50]. AOD at 0.86 μm is retrieved by matching the observed and calculated TOA reflentances, by a linear combination of fine and coarse aerosol modes, for selected channels. For other wavelengths, the AOD is determined by using the spectral dependence of aerosol optical properties unique to the aerosol model, included in the LUT. More detailed description of the algorithm can be found in [13].

VIIRS clouds mask (VCM) and internal tests are implemented to screen out pixels unfavorable for retrieval, e.g., cirrus, snow, ice, strong inhomogeneity, etc., and designate data quality. With respect to smoke conditions, an internal test is used to identify heavy aerosol based on the observed evidence that the Rayleigh-scattering-corrected TOA reflectance is usually concave lower in the band centered at 445 nm for heavy smoke and dust, owing to the enhanced absorption at 412 nm [13]. It's noteworthy that the upper limit of AOD retrievals at 550 nm is set to 5.0, because when the known aerosol optical properties for limited number of aerosol types (urban, generic, dust, smoke) are tested in the retrieval algorithm, the sensitivity of satellite measured reflectance to aerosol loading changes sign when AOD > 5.0; instead of increasing, TOA reflectance decreases due to absorption of radiation. This is not the case for clouds whose reflectance increases monotonically with optical depth.

Validation of the algorithm shows small positive bias of AOD at 550 nm over land (0.02) and ocean (0.03) for high-quality data, and the precision over water is about twice of that over land [13], comparable to the results of MODIS data [14].

2.1.4. Main Differences of the AOD Products

The AOD products evaluated in this work are based on different satellite sensors and algorithms, which means that the spatiotemporal sampling difference of sensors (due to orbit, swath width, footprint resolution), viewing geometry, sensor capability, data calibration, and algorithm differences (including cloud filtering, smoke masking, aerosol models, etc.) may contribute to the differences of their retrievals. Among these factors, differences in characterization of surface reflectance and aerosol models are the main sources of diversity in AOD retrievals over biomass burning regions [29]. VIIRS algorithm uses swath-based processing, thus the satellite footprint and its location are orbit dependent, making it difficult to characterize the surface BRDF. In addition, the VIIRS algorithm relies on prescribed spectral reflectance ratios and reflectance database over land [13,18]. The lack of accurate surface reflectance becomes a major issue over bright surfaces and when the sensitivity of TOA radiance to aerosols decreases [21]. In comparison, the MAIAC algorithm uses a fixed 1 km grid to combine data for up to 16 days within a sliding window. It uses time series analysis to separate atmospheric and surface characteristics with minimal assumptions and derive BRDF and SRC from multi-angle observations from different orbits [21]. The fixed (gridded) surface representation and characterization of surface reflectance using data over time helps to increase the accuracy of cloud and snow detection, atmospheric correction, and AOD retrievals, especially over bright surfaces [22]. Another important feature is the smoke masks used in both algorithms, which help to reduce cloud contamination commission errors [21,29].

*2.2. Aerosol Robotic Network (AERONET)*

AERONET is a ground-based federated network of globally distributed sun photometers to provide measurements of aerosol optical and physical properties [37], which have been widely used in characterizing aerosols and validating satellite retrievals. The AODs are available from direct measurements at multiple wavelengths (340, 380, 440, 500, 675, 870, and 1020 nm). In this work, the AERONET Version 3, Level 2.0 (cloud-cleared and quality-assured) [37,38] spectral AOD dataset from a total of 41 sites in September 2020 is used. The absolute uncertainty in the mid-visible band is about 0.01 [51]. The analysis region and geographical distribution of the AERONET sites are shown in Figure 1. As the evaluation is implemented for AOD at 550 nm, for consistency, the AERONET AOD at 550 nm is converted from the data at 500 nm using the Ångström exponent measured simultaneously for 440 to 675 nm. Additionally, AERONET AOD is confined according to the upper limit of the satellite AOD before the comparisons are performed, which means that any AERONET AOD higher than that upper limit is assigned to the limit value. For the MAIAC AOD, the upper limit is 4.0 and 6.0 at 470 nm for C6.0 and C6.1, respectively. Thus, we calculate the upper limit at 550 nm using the Ångström exponent for 440 to 675 nm. For VIIRS data, the AERONET AOD at 550 nm is confined to be less than or equal to 5.0.

*2.3. Collocation Method and Evaluation Metrics*

The comparison of satellite AOD products against AERONET data is performed for collocated records independently for each dataset. Considering the different spatiotemporal representations of satellite and ground-based AOD records, and following previous validation studies (e.g., [23,52]), the matchups are derived with the following requirements: (1) at least 100 (170) best quality retrieval pixels for MAIAC (VIIRS) retrievals at their native resolution, within a circle of 12.5 km radius centered on the ground stations, and (2) at least two ground measurements within a 30 min time window centered at the satellite overpass time. The limit for the number of valid pixels corresponds to 20% of the maximum potential retrievals in a circle. Spatial and temporal averages are performed to get the collocated measurements. In this work, only the MAIAC C6.0 and C6.1 data diagnosed as the best quality are used, which means both the cloud mask and cloud adjacency flags suggest "clear". For VIIRS data, we firstly used only best quality retrievals and found that some data under heavy smoke can be excluded and identified as low- or medium-quality.

Therefore, we also evaluated the VIIRS data by retaining retrievals of all quality levels that passed the internal heavy aerosol test with AOD > 0.5.

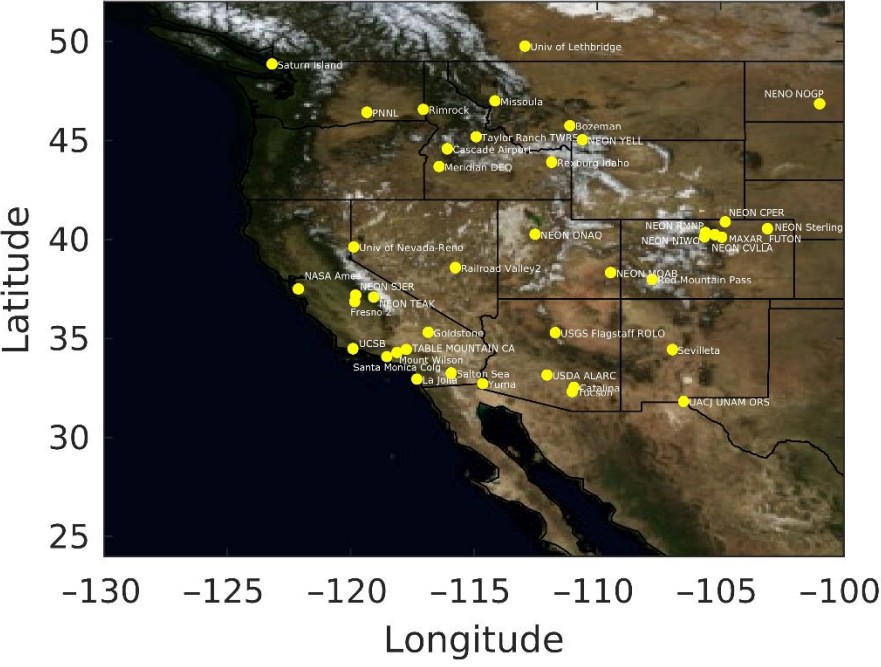

**Figure 1.** Map of the analysis domain and locations of the 41 AERONET sites (yellow dots). The names of these sites are labeled next to each location.

Statistical metrics used include mean bias (MB), root-mean-square error (RMSE), normalized mean bias (NMB), normalized mean error (NME), and Pearson correlation coefficient (*r*), which are calculated as follows:

$$MB = \frac{1}{N} \sum_i (m_i - o_i) \tag{1}$$

$$RMSE = \sqrt{\frac{1}{N} \sum_i (m_i - o_i)^2} \tag{2}$$

$$NMB = \frac{\sum_i (m_i - o_i)}{\sum_i o_i} \times 100\% \tag{3}$$

$$NME = \frac{\sum_i |m_i - o_i|}{\sum_i o_i} \times 100\% \tag{4}$$

$$r = \frac{\sum_i [(m_i - \overline{m}) \times (o_i - \overline{o})]}{\sqrt{\sum_i (m_i - \overline{m})^2 \times \sum_i (o_i - \overline{o})^2}} \tag{5}$$

Here, the subscript *i* represents the pairing of N AERONET observations (*o*) and satellite data (*m*).

In addition, the percentage of data falling within the expected EE of $\pm(0.05 + 15\% \times AOD)$ over land established by MODIS (%EE) [18,53] is evaluated. This form of EE has been widely used in AOD validation studies, which corresponds to the spread of normally expected errors for operational MODIS AOD with 66% of data points lying within the EE [18,27]. Thus, a higher percentage of points falling within the EE suggests a better performance.

Given the possible sampling differences among the sensors, we also perform a statistical validation of the simultaneously collocated retrievals of all three products with AERONET, which allows the AODs to be intercompared under the same conditions. For each surface site, the satellite-AERONET matchups are derived using the requirements

described above, and a further filter is applied to only keep the data of the three satellite sensors overpassing within 30 min for temporal collocation.

### 2.4. Daily Gridding of MAIAC C6.1 and VIIRS

Considering the approximate equatorial overpassing time of NOAA-20 and Aqua, we compare the AOD retrievals from Aqua MODIS MAIAC and NOAA-20 VIIRS for daily gridded data, to provide more insight into data coverage, AOD distribution, and cloud screening (see method in Section 2.5). Aqua and NOAA-20 satellites operate at different orbit altitude and inclination, and a common scene in Aqua MODIS and NOAA-20 VIIRS can occur across different data granule times. The approximate collocation phasing of the two satellites is 3 days [54]. Thus, to reduce the impact of temporal sampling difference, rather than comparing the AOD retrievals directly for spatially collocated pixels, we compare the daily re-mapped AOD at a 0.1° grid. The grid resolution is chosen to be comparable to the collocation and spatial averaging (12.5 km) applied in the evaluation against AERONET data. The best-quality retrievals are filtered, which are confidently clear and not adjacent to a cloud pixel. Any valid retrievals whose center geolocation fall within the same grid box are averaged to get the daily gridded data. The intercomparison of AOD products extends the evaluation by providing spatial context of their performance [30].

### 2.5. CALIPSO

CALIOP is an active lidar on board the CALIPSO satellite, launched into orbit as part of the A-Train satellites (including Aqua) in April 2006. On 20 September 2018 CALIPSO executed a series of maneuvers to join CloudSat's orbit to become another one of the C-Train satellites ("C" for CALIPSO and CloudSat) at the orbit altitude 16.5 km below A-train. CALIPSO intersects the A-Train ground track about every 20 days. It detects vertical profiles of attenuated backscatter (523 nm and 1064 nm) with the along-track resolution of 333 m and vertical resolution of 30 m below 8.5 km [55]. Products of the number and extent of layer features of aerosol or cloud are developed using the backscatter profiles [56]. The Level 2 aerosol and cloud products are provided at the spatial scales at 1/3, 1, 5, and 40 km regarding different data averaging and detectability.

In this work, the Version 4.20 Lidar Level 2 Vertical Feature Mask, 1 km Cloud layer, and 5 km Aerosol layer products are used to evaluate the MAIAC C6.1 and VIIRS cloud masks. Collocated observations of each dataset and CALIOP are derived separately at their native resolutions, requiring that the nearest pixel of AOD retrieval relative to a CALIOP profile is located with a distance less than 1 km, and sampled within ±10 min relative to the CALIOP observation time. For MAIAC and VIIRS, their cloud masks are converted to binary values, namely either "clear" or "cloudy". For CALIOP 1 km cloud layer data, the binary mask is derived by assigning the profiles with the "Number of Layers Found" reporting at least one cloud layer as "cloudy", and the remaining profiles as "clear". The CALIOP aerosol layer data is used to filter cloud masks under smoke conditions (see Section 3.2.2). Based on the method reported in [25], the cloud mask matchups are classified into four categories: True Positive (TP), False Positive (FP), True Negative (TN), and False Negative (FN). The metrics of overall Accuracy, true positive rate (TPR), and true negative rate (TNR) are calculated as follows:

$$\text{Accuracy} = \frac{\text{TP} + \text{TN}}{\text{TP} + \text{TN} + \text{FP} + \text{FN}} \tag{6}$$

$$\text{TPR} = \frac{\text{TP}}{\text{TP} + \text{FN}} \tag{7}$$

$$\text{TNR} = \frac{\text{TN}}{\text{TN} + \text{FP}} \tag{8}$$

## 3. Results

*3.1. Validation of AOD at 550 nm against AERONET Data*

3.1.1. Individual Products

In this section, AODs from the three datasets are evaluated using the AERONET observations as the "ground truth". Figure 2a–d show scatterplots of the AOD matchups (satellite versus AERONET) for the individual products along with the EE of $\pm(0.05 \pm 15\%$ AOD) for the operational DT product over land [18,53]. The statistical metrics are listed in Table 2. The retrievals from all the four groups are generally well correlated with those of AERONET, with the correlation coefficient ranging from 0.90 to 0.94. MAIAC C6.0 is found to drastically underestimate AOD (Bias = $-0.104$, NMB = $-25.3\%$, NME = 35.0%). Compared to MAIAC C6.0, C6.1 exhibits a slight positive bias (Bias = 0.029, NMB = 6.8%, NME = 26.8%) and lower RMSE than C6.0, with about 73% of retrievals falling within the EE of $\pm(0.05 \pm 15\%$ AOD). This suggests improved retrieval accuracy related to the updated aerosol model and parameters developed for smoke conditions. Specifically, the dynamically increasing aerosol absorption and decreasing coarse mode fraction at AOD > 0.6 implemented in MAIAC C6.1 contribute to the higher AOD. In comparison, VIIRS shows a slightly higher positive bias than for MAIAC C6.1, which is within the technical requirement of the retrieval product and slightly larger than reported over land over global AERONET sites for multitype years [13]. Including VIIRS pixels of all quality when they are diagnosed as smoke leads to similar accuracy compared to the results for high-quality data only (Figure 2d). Notably, the total number of matchups for MAIAC C6.1 and C6.0 are about 2.12 and 2.32 times of that for VIIRS. This is because data from both Aqua and Terra MODIS are included. Even if we only consider a half of the data for MAIAC, the total number of matchups is still higher than that for VIIRS, which likely indicates the better capability of MAIAC to retrieve over the bright surfaces, since many of the AERONET sites are in arid areas. Similar results were reported in [25]. This can also be seen for the larger number of MAIAC retrievals over the month in our analysis for gridded AOD (see Section 3.2.1).

**Table 2.** Statistical measures of satellite AOD (550 nm) compared with the AERONET data for the 41 sites in western U.S. shown in Figure 1. Results are listed for the validation of each product (MODIS MAIAC C6.0, C6.1, and NOAA-20 VIIRS) individually, for collocated data when the three datasets have valid retrievals simultaneously, and merged data. The metrics shown here include the total number of points (N), Pearson correlation coefficient (*r*), mean bias (MB), root-mean-square error (RMSE), normalized mean bias (NMB) and normalized mean error (NME).

| Product Name | N | *r* | RMSE | Bias | NME (%) | NMB (%) |
|---|---|---|---|---|---|---|
| Individual Product/Collocated | | | | | | |
| MAIAC C6.0 | 1568/343 | 0.932/0.941 | 0.357/0.254 | $-0.104/-0.074$ | 35.01/30.68 | $-25.32/-21.27$ |
| MAIAC C6.1 | 1714/343 | 0.939/0.947 | 0.252/0.187 | 0.029/0.003 | 26.81/24.20 | 6.76/0.74 |
| VIIRS | 738/343 | 0.910/0.946 | 0.286/0.225 | 0.058/0.059 | 35.51/33.69 | 14.07/16.87 |
| VIIRS (retain smoke) | 843/— | 0.900/— | 0.370/— | 0.083/— | 36.40/— | 15.42/— |
| Merged (average of collocated data) | | | | | | |
| MAIAC C6.1 + VIIRS | 343 | 0.961 | 0.175 | 0.031 | 24.74 | 8.82 |

The dependence of AOD retrieval bias against AERONET AOD is analyzed by dividing all the data matchups into 11 bins (Figure 2e–l). Overall, MAIAC C6.0 shows an increasingly negative bias when AERONET AOD increases, suggesting a stronger negative bias towards thicker smoke. In comparison, for the first 10 bins (AOD < 3.2), the median biases of MAIAC C6.1 are around zero. VIIRS data tends to be positively biased over the bins from 0.53 to 3.2. A weak trend of increased bias of VIIRS EPS AOD at higher levels is also reported over other regions (e.g., [57]). Notably, over very thick smoke with AERONET AOD > 3.2, both MAIAC C6.1 and VIIRS have strong negative biases. However, due to the very limited number of matchups in that bin, this result needs to be further validated. In

addition, the spread of retrieval bias becomes generally larger when the AERONET AOD increases, suggesting the greater uncertainty under high aerosol loading. On country, the relative bias tends to become smaller, which may be related to the limited sample size and the hyperbolic behavior of relative metrics.

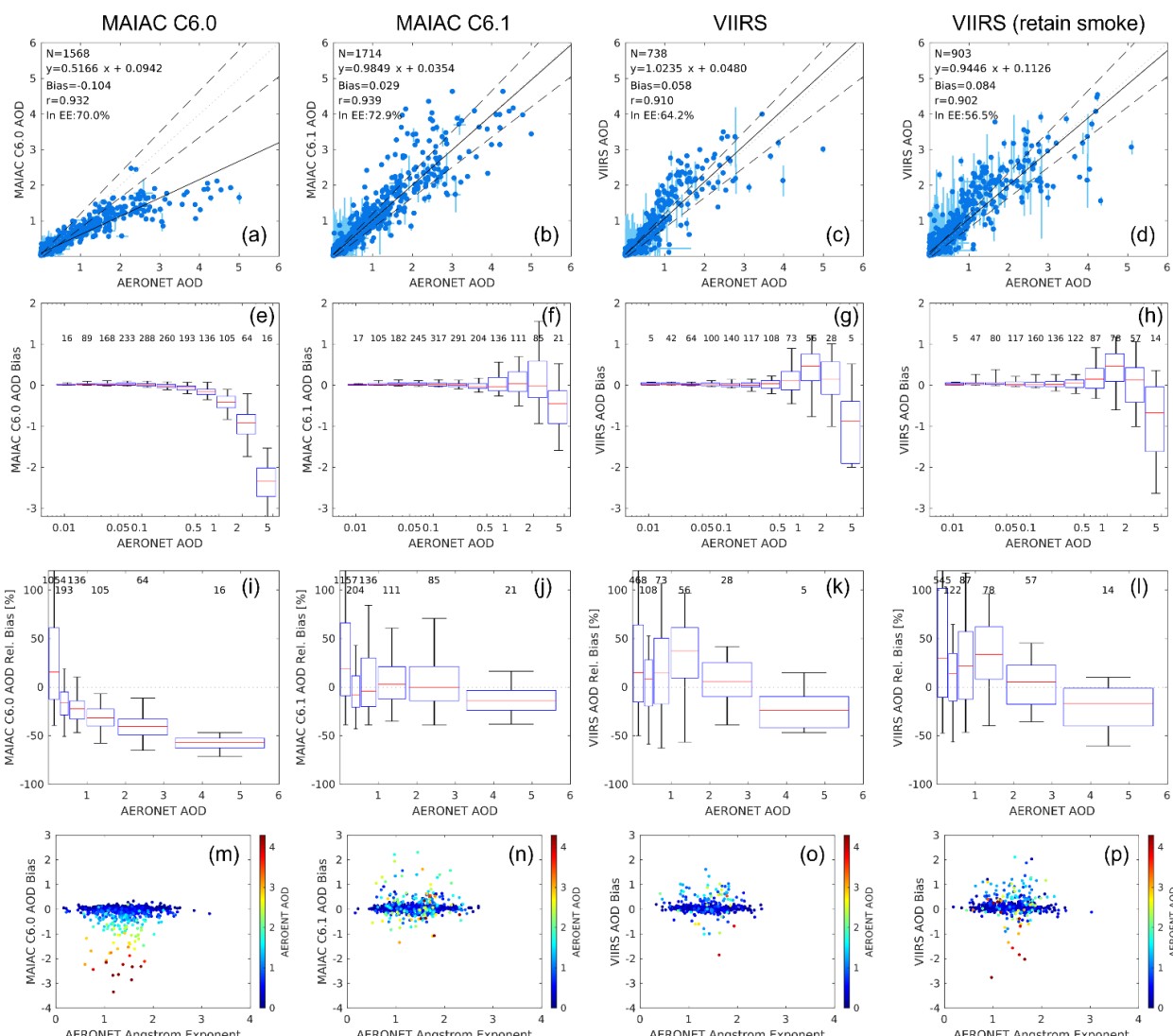

**Figure 2.** Comparison of three satellite AOD products of MODIS MAIAC C6.0, MAIAC C6.1, and VIIRS (best-quality and retaining all-quality retrievals passing the internal heavy aerosol test with AOD > 0.5) against AERONET measurements over the western U.S. in September 2020. (**a–d**) Scatter plots comparing the AOD (550 nm) retrievals. The horizontal and vertical bars represent the ±1σ variability in the corresponding space and time window for each satellite-AERONET matchup. The grey dotted line represents the 1:1 line. Statistical measures of the validation are noted within each plot. The black dashed lines are the error envelope (EE) of ±(0.05 + 15% AOD). (**e–h**) Box-whisker plots of the satellite AOD retrieval bias as a function of AERONET AOD (shown for 11 bins). The bin sizes are set in logarithmic scale. For each AOD bin, the red line represents the median, the upper and lower bounds of the box are the 75th and 25th percentiles, and the ends of vertical lines are the 5th and 95th percentiles. The number of matchups in each bin is noted at the top. (**i–l**) Similar as (**e–h**) but for relative bias (%). The bin setting follows (**e–h**) but the lower six bins are merged (0–0.29). (**m–p**) Scatter plots of the satellite AOD bias as a function of Ångström Exponent (440–675 nm) measured at the AERONET sites. Data are color coded by the AERONET AOD.

To elucidate the dependence of the retrieval errors to aerosol type, we examine the variation of the AOD bias against the AERONET Ångström exponent (AE), which represents the spectral dependence of AOD and is often used as a proxy of particle size [58]. It also provides information of the potential aerosol type, e.g., dust aerosols are typically in coarse mode, and biomass burning aerosols and secondary organic aerosols are mostly in fine mode. Figure 2m–p present scatterplots of AOD retrieval bias versus AERONET AE, color coded according to the AERONET AOD. For MAIAC C6.0, the significant negative bias tends to correspond to a high loading of finer particles (Figure 2m), confirming the limited performance of C6.0 for cases of thick smoke. However, the biases for MAIAC C6.1 and VIIRS do not seem to exhibit a dependence on particle size. The AE values range from about 0.5 to 2.5, and a slightly smaller spread of AE for VIIRS than for MAIAC C6.1 is shown.

### 3.1.2. Collocated Retrievals for All Three Products

As the validation for individual products may not be ideally comparable due to sampling difference, we further filtered the data of simultaneous (here within 30 min) collocation with AERONET among the three products. Note that only MAIAC data from Aqua MOIDS are kept by applying this filter. The results in Figure 3 and statistical metrics (Table 2) overall affirms the performance of the individual products.

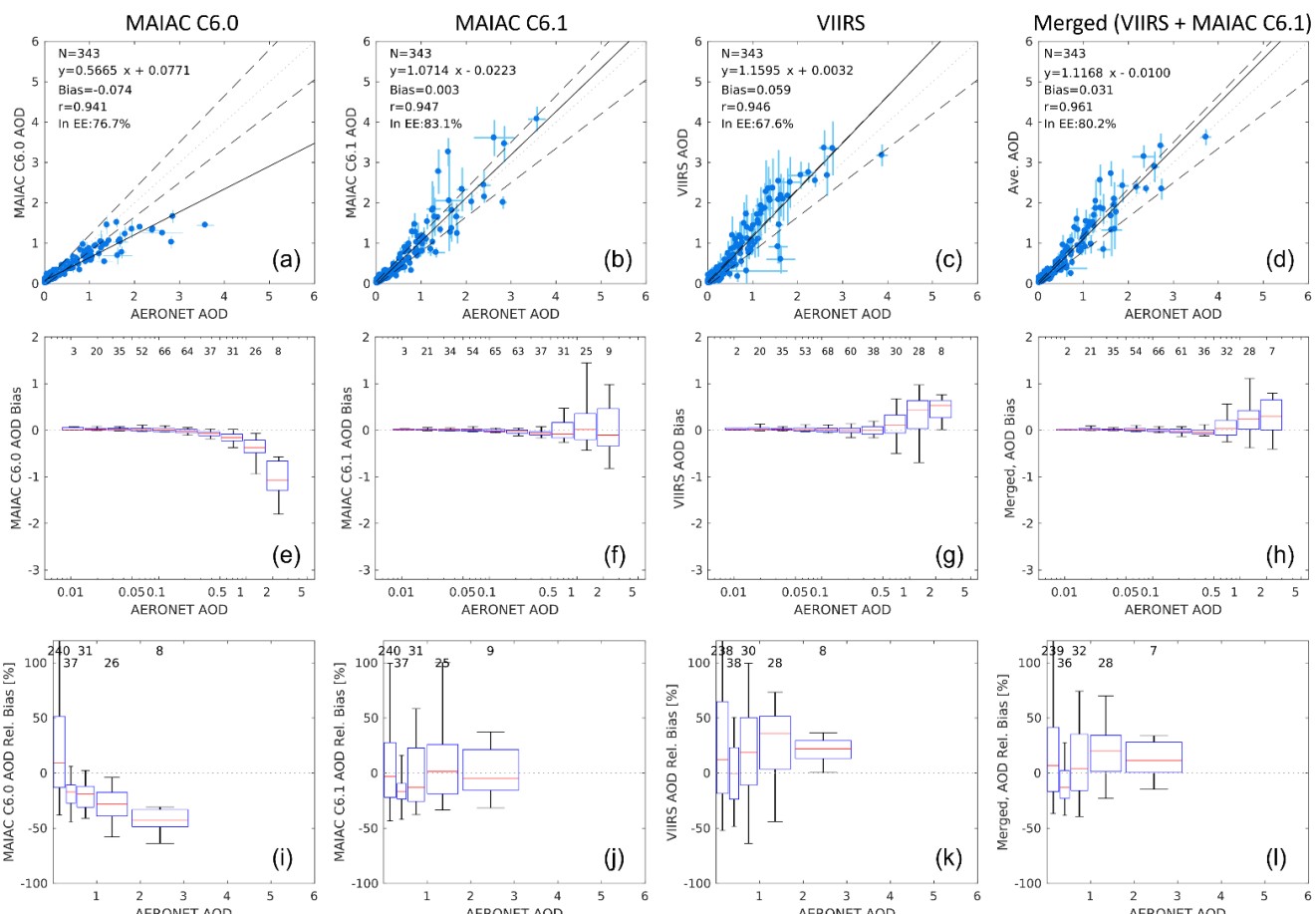

**Figure 3.** Similar as panels (**a**–**l**) in Figure 2, but for best-quality satellite AOD and AERONET AOD matchups when all the three products are available simultaneously. The first three columns are results for MAIAC C6.0, MAIAC C6.1, and VIIRS. The last column corresponds to validation of merged AOD by averaging the matchups for VIIRS and MAIAC C6.1.

Based on the collocated retrievals from MAIAC C6.1 and VIIRS, we attempt to merge them by averaging those matchups, and validation of the merged AOD is also presented

in Figure 3. In comparison with any individual dataset, the merged retrievals show lower RMSE and improved correlation with the AERONET data, and the bias is in between of the two individual products. Thus we can expect that, when bias-corrected, the blended dataset using collocated retrievals from MODIS MAIAC C6.1 and VIIRS has the potential to provide AOD with a higher accuracy than the individual products.

### 3.2. Comparison of MAIAC C6.1 and VIIRS

3.2.1. Comparison of Gridded AOD Maps

Gridded AODs on 0.1° maps are first compared directly on a monthly scale to examine differences in spatial distributions and number of retrievals. The grid resolution is chosen to make the retrieval performance of gridded data most comparable with those derived against AERONET based on our collocation method (Section 2.3). Nevertheless, for the monthly average data, we note that the average map is equivalent to level-3 data which may have different statistics than the validation shown in the above section. Figure 4a–e display the results for Aqua/Terra MODIS MAIAC C6.0, C6.1, and VIIRS. MAIAC exhibits a larger number of retrievals at the native resolution than VIIRS in some regions, for which the surface reflectance may be an important factor. Over bright surfaces and sparsely vegetated areas in the western U.S., relatively less data availability of VIIRS has been reported [25]. Meanwhile, there is a greater number of retrievals from Terra MODIS than from Aqua MODIS, which is likely related to the frequency of cloudy conditions in the afternoon. Note that this difference may not be valid in other months due to seasonal variation of cloud coverage. All these products show complete spatial coverage when averaged over the month. VIIRS exhibits improved data availability than from a previous version of algorithm as evaluated in [25].

Considering the approximate overpassing time of NOAA-20 and Aqua, we compare the AOD distributions from Aqua MODIS MAIAC and NOAA-20 VIIRS. The monthly maps of Aqua MODIS MAIAC C6.1 and VIIRS show very similar spatial distribution patterns, albeit with differences in magnitude (Figure 4). Owing to the differences among platforms, sensor specifications, and algorithms, etc., there are grid cells where the retrieval is valid for one sensor but absent for the other. For this reason, we intercompare the monthly average AOD requiring collocation, namely by filtering only the daily grid cells where both products are valid. Compared to Aqua MAIAC C6.0, C6.1 shows increased AOD values by 0.027 over the domain (Figure 4f). The increase is especially noticeable over the areas affected by thick smoke plume, consistent with the validation based on the AERONET data. Meanwhile, VIIRS AOD tends to be slightly higher than MAIAC C6.1 by 0.036 (Figure 4g), showing different trends over the ocean and land surface. Over ocean, VIIRS is generally lower than Aqua MAIAC C6.1, except for the areas near the northwest coast impacted by smoke plumes, where the differences are more scattered. Over land, the difference is most noticeable in the northern portion of this domain, which is the major regional transport passage of the wildfire smoke in September 2020. Meanwhile, the difference between VIIRS and Aqua MAIAC C6.1 is more varied and scattered in California and some areas in Mexico, indicating a larger spread of the retrieval difference over thick smoke and heterogeneous terrain.

Time series of the daily domain average AODs for the spatially collocated grid cells with valid data are shown in Figure 5a (right *y*-axis), to present temporal evolution of AOD differences. The result indicates larger estimates for VIIRS compared to MAIAC C6.1 nearly over the whole month, with larger deviations during the most severe days (11–20 September) which correspond to the higher numbers of grid cells with AOD > 1.5 (Figure 5a, left *y*-axis). These differences are in accordance with the VIIIRS AOD validation against AERONET data. Figure 5b shows the frequency distributions of the collocated gridded AODs for the two products, which suggests very similar patterns. The distribution for VIIRS tends to be narrower and tilted to the lower end; the peak around 2.0 likely contributes to the larger domain average AOD.

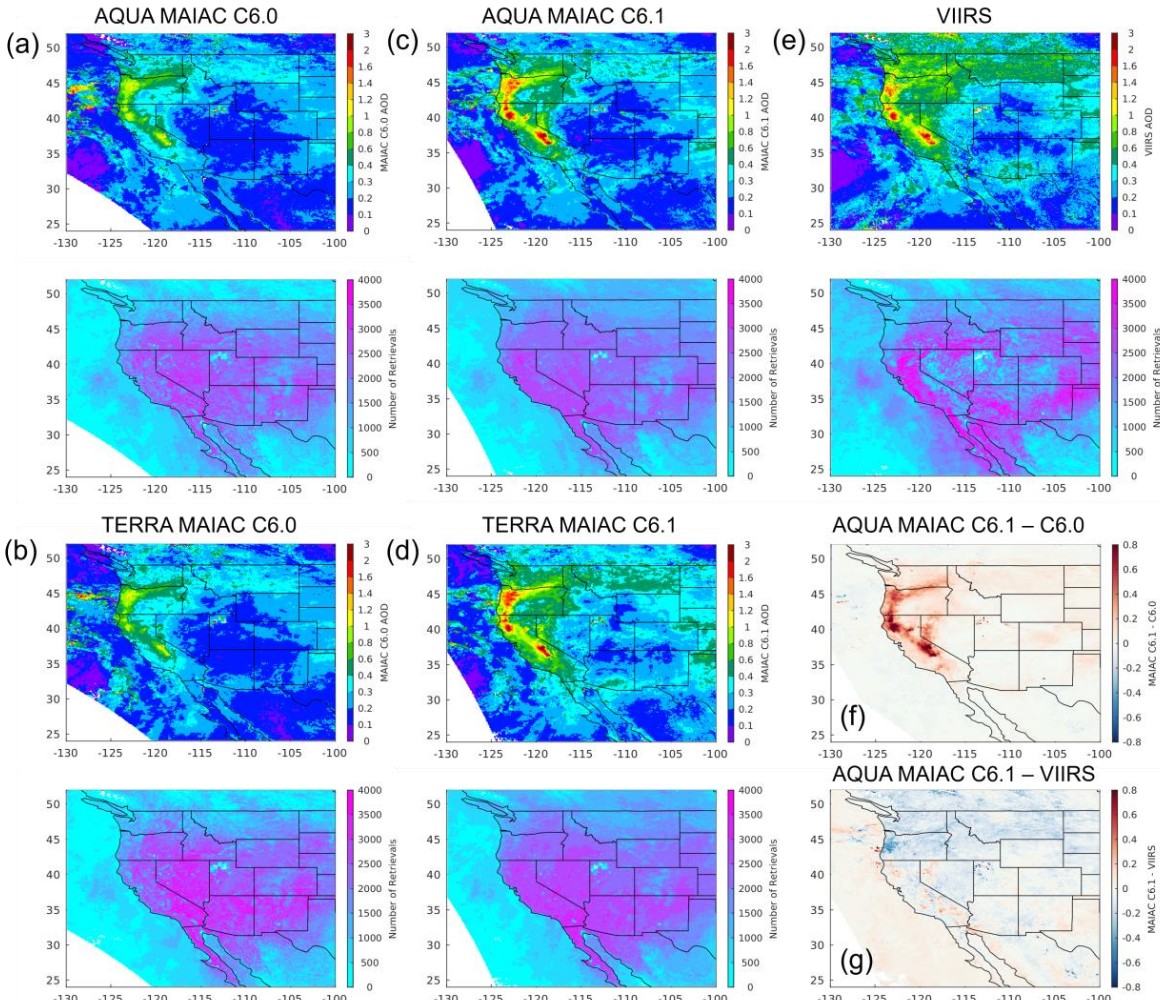

**Figure 4.** Maps of monthly average AOD (550 nm) and number of high-quality retrievals in the 0.1°
resolution grid from Aqua and Terra MAIAC C6.0 (**a**,**b**), C6.1 (**c**,**d**), and VIIRS (**e**). The portion of
missing data in MAIAC in the lower left corner of the plotting domain is due to the spatial extent of
MAIAC data over North America. Panels (**f**,**g**) are the difference of monthly average AODs between
(**f**) Aqua MAIAC C6.1 versus C6.0, and (**g**) Aqua MAIAC C6.1 versus VIIRS. The monthly AOD is
averaged using daily collocated grid cells where both products have retrievals simultaneously. The
domain averages of (**f**,**g**) are 0.027 and −0.036, respectively.

Due to the differences in sampling, pixel size (VIIRS has a slower growth of the footprint
size with the scan angle than MODIS), sensor characteristics, and retrieval algorithms, besides
the collocated daily gridded AODs, there are also non-collocated grid cells that contain
valid retrievals only from one product. Examining these different matching groups may
provide more insights into the retrieval performance. The number of collocated daily grid
cells ($1.27 \times 10^6$) account for 92% of the total grid cells with valid daily AODs from Aqua
MODIS MAIAC C6.1 ($1.38 \times 10^6$), and 82% of those from VIIRS ($1.55 \times 10^6$) (Figure 5b). The
total number of valid data suggests a higher availability of VIIRS compared with MAIAC
at gridded level over the analysis domain, in contrast to the larger number of matchups for
MAIAC when compared with AERONET (Section 3.1). There are multiple possible reasons:
(1) sampling differences leading to distinct retrieval conditions that the clouds may have
moved out of the field of view when VIIRS overpassed, and vice versa; (2) the finer spatial
resolution of VIIRS pixels resulting in more chances of capturing clear scenes at gridded level,
when the clouds are scattered; and (3) the wider swath of VIIRS giving more valid retrievals.
In addition, the higher coverage of VIIRS AOD maps could be contributed by cloud masking
method, if the MAIAC cloud mask tends to mistakenly exclude clear pixels as cloudy or VIIRS

tends to misclassify cloudy pixels as clear. This is further discussed in the evaluation of cloud masks using CALIOP data (Section 3.2.2).

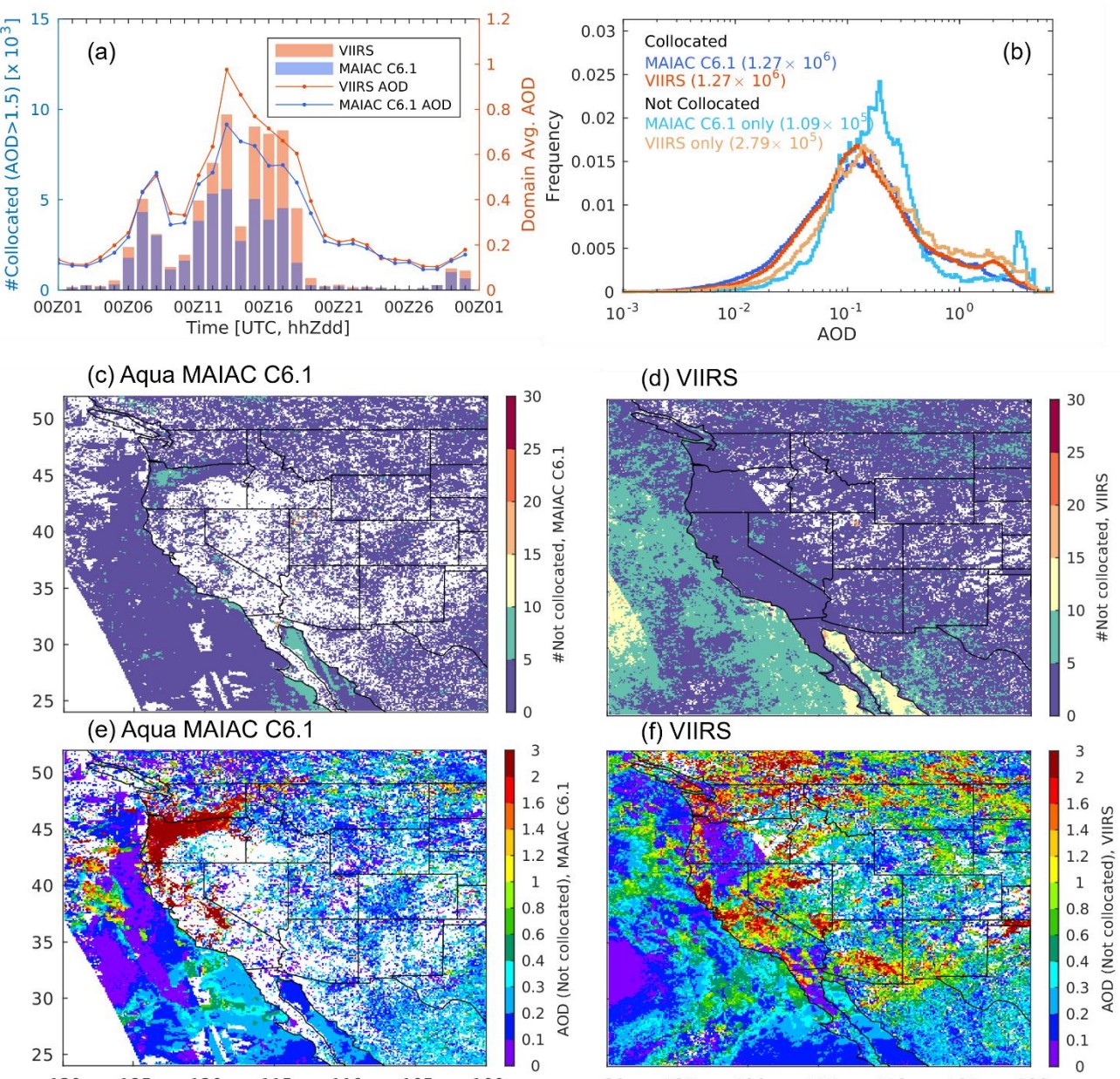

**Figure 5.** (**a**) Daily time series of number of grid cells with AOD > 1.5 (left y-axis, bars) and domain average AOD (right y-axis, dotted lines) for collocated gridded data for Aqua MODIS MAIAC C6.1 and VIIRS; (**b**) Frequency distribution of the AOD values in bins in logarithmic scale for collocated data from VIIRS, and Aqua MODIS MAIAC C6.1, and the not collocated data individually; (**c,d**) Map distribution of the number of days with non-collocated data on the 0.1° grid in September 2020 and (**e,f**) the monthly average AOD across those non-collocated grid cells for Aqua MODIS MAIAC C6.1 and VIIRS, respectively.

To interpret the non-collocations, we first compare the monthly spatial distribution of the total occurrence of non-collocated grid cells and their average AOD (Figure 5c–f). Over ocean, more non-collocations happen for VIIRS, contributing to the greater data coverage shown previously (Figure 4). Over land, the non-collations in MAIAC C6.1 are most discernable over areas affected by smoke originating from Oregon and California with AOD values above 2.0 (Figure 5c,e), which is also shown by the peak in frequency

distribution around AOD of 3.5 for non-collocated grid cells of MAIAC C6.1 (Figure 5b). The absence of VIIRS AOD is largely due to the missing retrievals over thick plumes, which turns out to be mostly related to the retrievals over heavy smoke that are classified as medium and low quality, owing to being larger than the upper limit (AOD = 5.0), or internally masked as cloud, adjacency to cloudy pixel, or inhomogeneity. This will be discussed in Section 3.3. Additionally, the non-collocations of MAIAC C6.1 that occurred on less than 5 days during the month are spread over the domain but correspond to small AOD (<0.4), therefore, they might be related to the sampling difference.

By contrast, the non-collocations for VIIRS that existed on more than 5 days are mostly in Mexico and high latitudes (Figure 5d,f), with the average AOD at these non-collocated grid cells larger than 0.4 and even higher. Additionally, the frequency distribution for non-collocated VIIRS data is shifted higher than that for the collocated data. These high AODs are more likely attributed to cloud masking, besides sampling and resolution differences, either due to clear smoke pixels mistakenly filtered as clouds by MAIAC, or residual cloudy or cloud-adjacent pixels that pass the cloud screening process in VIIRS but are excluded by MAIAC. Under cloudy conditions or near clouds, retrievals can be biased high due to scattered photons from cloud sides [15,59], hydrated aerosols, cloud fragments, and dissipating clouds [60], which all enhance AOD albeit they are not true cloud contamination. In the following sections, we implement an evaluation of the cloud masking based on CALIOP data (Section 3.2.2) and case analysis (Section 3.3) to illustrate the possible contributors of the non-collocations.

### 3.2.2. Cloud Masking

To interpret the possible contribution by cloud masking to the daily data coverage of MAIAC C6.1 and VIIRS, we compare the cloud identifications from the two datasets against the CALIOP data (Table 3). The total numbers of observation matchups with CALIOP for MAIAC C6.1 and VIIRS are 26,241 and 32,865, respectively. The overall accuracy of cloud masking of the two datasets are similar (Table 3), with VIIRS showing a slightly higher value (87.4%). Meanwhile, the TPR of MAIAC (95.4%) is higher than that for VIIRS (85.6%), meaning a higher possibility of VIIRS (14.4%) compared to MAIAC (4.6%) to incorrectly designate cloudy conditions as clear, which may lead to cloud leakage and contaminations in AOD retrievals, similar as the results reported in the literature [25,59]. On the other hand, the TNR for MAIAC (78.6%) and VIIRS (88.2%) suggest that about 22% and 11% of the clear pixels identified by CALIOP are masked out as cloudy for MAIAC and VIIRS. Overall, VIIRS tends to show a better balance between errors due to false alarm and cloud leakage, and MAIAC has better performance in terms of removing cloudy scenes. MAIAC tends to be conservative in determining which pixels are cloud free, which is supposed to be due to the block-level characterization of surface reflectance, as well as sampling difference compared with CALIOP [25]. This helps to explain the larger AOD coverage of VIIRS than for MAIAC as shown in Section 3.2.1.

To further examine the accuracy of cloud masks under smoke conditions, we filter the smoke-covered and clear scenes based on CALIOP data following the three criteria: (1) At least one aerosol layer is detected; (2) No cloud layer; (3) Integrated aerosol optical depth is above 0.3. This filtering yields 1839 matchups between CALIOP and VIIRS data and 1403 for MAIAC C6.1. The percentage of smoke-covered scenes being correctly diagnosed as "clear" is higher for VIIRS (90.7% vs. 77.4%) (Table 3), in consistency with the general performance of cloud masks under all clear scenes. Thus, over smoke plumes, the accuracy of VIIRS cloud masking tends to be better than that for MAIAC C6.1. This is in contrast with the data coverage for the gridded data (Figure 5), which suggests fewer valid retrievals of VIIRS over thick smoke plumes than Aqua MAIAC C6.1. Therefore, cloud masking does not play a substantial role for the missing best-quality AOD retrievals from VIIRS over areas of heavy smoke. Other limits and tests in the algorithms may be more important, which will be discussed in the case studies in the next section.

**Table 3.** Confusion matrix for the comparison of pixels from each cloud mask associated with the two AOD products against collocated information from the CALIOP cloud layer data. The abbreviations in parentheses denote (for both sets of data): true positive (TP), false positive (FP), true negative (TN), false negative (FN), true positive rate (TPR), and true negative rate (TNR).

| | | **VIIRS** | | | **MAIAC C6.1** | | |
|---|---|---|---|---|---|---|---|
| | | **Cloudy** | **Clear** | | **Cloudy** | **Clear** | |
| CALIOP | Cloudy | 8807 (TP) | 1485 (FN) | 85.6% (TPR) | 9121 | 444 | 95.4% |
| | Clear | 2661 (FP) | 19,912 (TN) | 88.2% (TNR) | 5190 | 19024 | 78.6% |
| Accuracy | | 87.4% | | | 83.3% | | |
| | | VIIRS | | | MAIAC C6.1 | | |
| | | Cloudy | Clear | | Cloudy | Clear | |
| CALIOP | Smoke | 172 | 1667 | 90.7% (TNR) | 317 | 1086 | 77.4% |

*3.3. Case Studies*

In this section, we analyze cases in September 2020 to illustrate and validate the findings shown in the above sections. Several observation days and AERONET stations near the fire sources (NEON TEAK) and remotely affected by transported smoke (NEON NOGP and USDA ALARC) are selected. The non-collocations between MAIAC C6.1 and VIIRS AODs are analyzed to elaborate the discrepancy in their retrievabilities.

3.3.1. NEON TEAK, 14 September 2020

The AERONET station NEON TEAK (37.01°N, 119.01°W) is close to the wildfires in central California during September 2020. As seen in Figure 6, the AOD observed from sun photometer at this site shows enhancements on 6–17 September due to the smoke. Similar variation in surface $PM_{2.5}$ mass concentrations at a nearby surface monitor (34.3 km) confirms the adjacency to fire emissions upwind and smoke near the land surface. During this period, the AODs from Aqua MODIS MAIAC C6.1 generally present a correlated variation trend with AERONET data. By contrast, the VIIRS data at this site is almost missing, meaning that the high-quality retrievals are too sparse to populate valid matchups. The spatial distributions of the high-quality VIIRS AOD on 14 September (Figure 6d) present less data around the NEON TEAK site. The missing data are also seen for the smoke plumes originating from north California and Oregon.

The retrieval quality of VIIRS data is examined over optically thick smoke near the NEON TEAK site. Figure 7 shows the map distributions of VIIRS AODs with different quality levels (QCAll = 0: high, 1: medium, 2: low), their quality level flag, and the external/internal masks for clear, cloudy, and heavy aerosol conditions. By including all available data, VIIRS exhibit similar coverage as MAIAC C6.1 with higher magnitude over the heavy smoke areas. The pixels of thick smoke designated as cloudy by the external cloud mask (Figure 7c) are called-back by the internal heavy aerosol mask (Figure 7d) to obtain retrievals. Some of these retrievals are labeled as low quality (QCAll = 2, Figure 7b) because of exceeding the upper limit of the valid range (AOD = 5.0). In addition, some pixels are designated as medium quality (QCAll = 1) for the surrounding thinner smoke that fail the internal tests of cloud shadow, cloud, cloud adjacency, or inhomogeneity. Therefore, improvements of the algorithm and quality filtering are necessary to effectively retain the retrievals over smoke plumes.

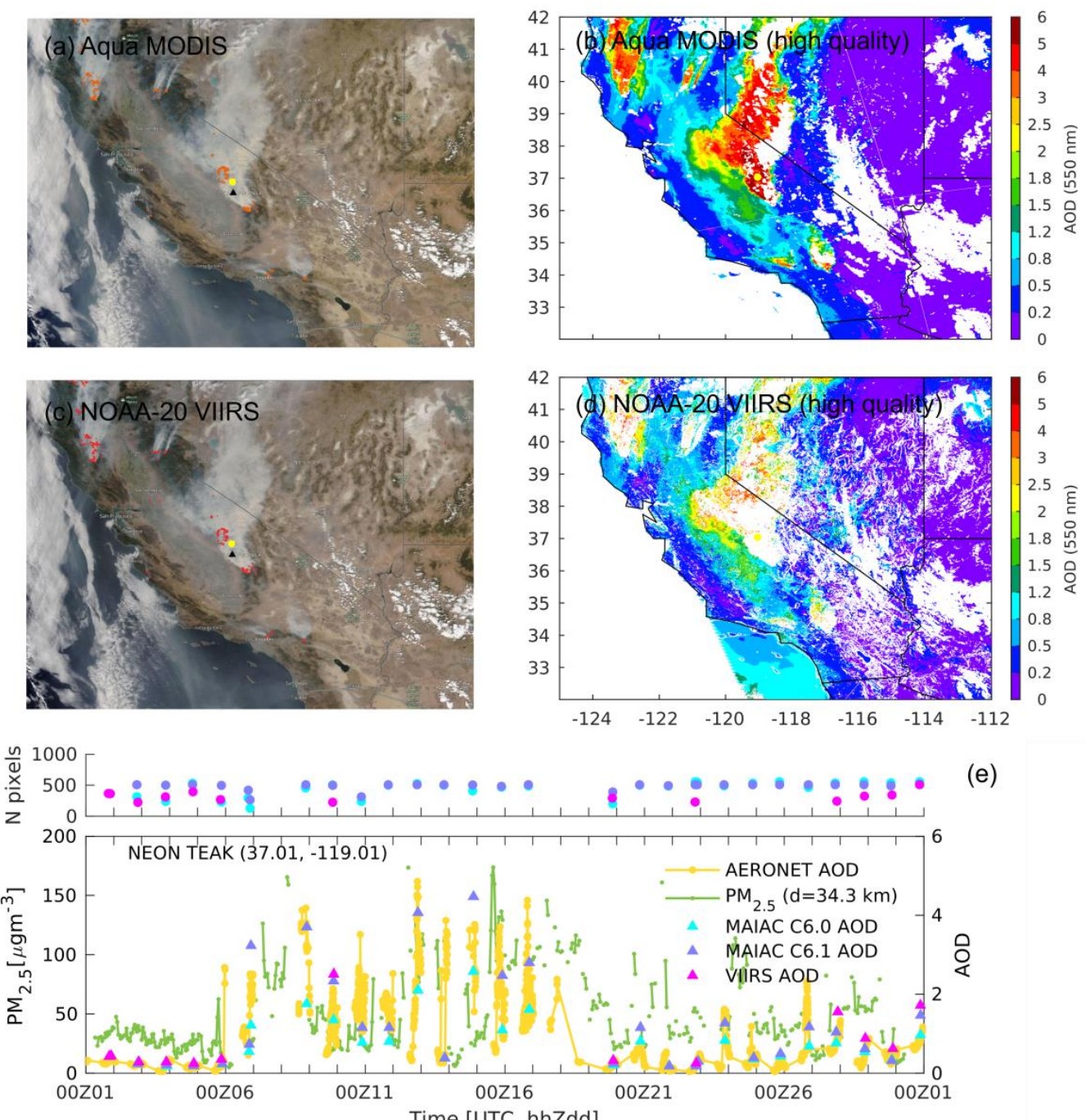

**Figure 6.** Snapshots of (**a**) Aqua MODIS and (**c**) NOAA-20 VIIRS true color images on 14 September 2020, obtained from the NASA Worldview (https://worldview.earthdata.nasa.gov/, accessed on 8 April 2022). The orange and red dots stand for fire hotspots detected by the two sensors. Panels (**b**,**d**) show map distributions of AOD (550 nm) from Aqua MODIS MAIAC C6.1 and VIIRS on the same day. The data (high-quality) are shown at their native resolution. (**e**) Time series of AOD (right axis) at the AERONET site NEON TEAK (yellow dot) and collocated satellite AODs at this site from the three products in September 2020. The dots (in same color coding as the AOD plot) stand for number of satellite pixels filtered to derive the matchups with AERONET. The green line shows surface $PM_{2.5}$ concentration (left axis) measured at the closest EPA monitor (black triangle). The time format on x-axis is hhZdd (hour and day of month). Distance between the EPA monitor and NEON TEAK is noted in parenthesis (d = 34.3 km).

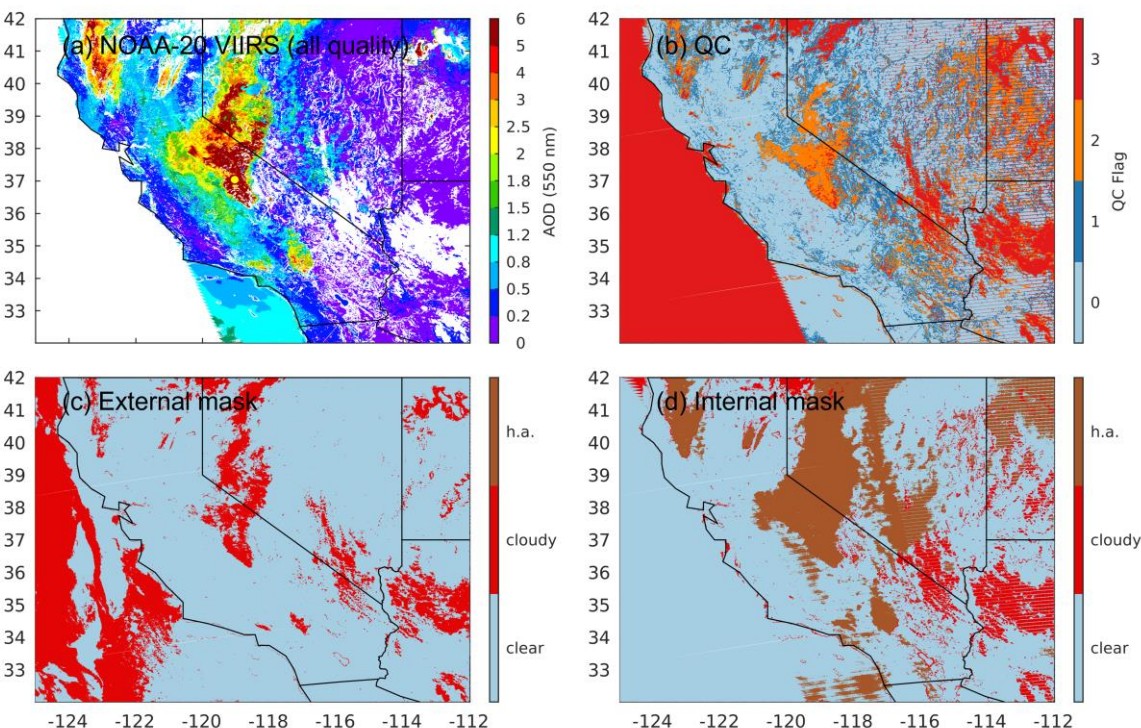

**Figure 7.** Maps of NOAA-20 VIIRS data on 14 September 2020. (**a**) AOD retrievals in all quality category (QCAll = 0, 1, 2); (**b**) Retrieval quality flag (QCAll, 0: high, 1: medium, 2: low, 3: no retrieval); (**c**) External mask and (**d**) internal mask for clear, cloudy, and heavy aerosol (h.a.) conditions.

### 3.3.2. NEON NOGP and USDA ALARC, 13 September 2020

The AOD enhancements associated with transported and aged smoke plumes are captured at NEON NOGP (46.77°N, 100.92°W) and USDA ALARC (33.08°N, 111.97°W) (see Figure 8a,c). On 13 September, scattered higher AOD values can be seen on the VIIRS map near the border of Mexico (Figure 8d). The greater departures between AOD from the two products over the area with scattered clouds suggests larger spread of uncertainty in the retrievals over clear scenes around cloudy areas. Meanwhile, over USDA ALARC (Figure 8e), satellite AOD is available for VIIRS but missing for both MAIAC C6.0 and C6.1, as shown in Figure 8b with the scattered aeras of missing data in the south of Arizona. The missing coverage of MAIAC is likely due to the more conservative cloud masks in the algorithm.

Over NEON NOGP, the enhanced AOD corresponding to smoke is observed on 13–15 September. Notably, the surface $PM_{2.5}$ does not increase simultaneously (Figure 9c), implying that the smoke is mostly lofted at high altitudes above the planetary boundary layer (PBL) and is not mixed down to the surface. The non-collocation of satellite retrievals from MAIAC C6.1 and VIIRS is found on 13 September. Comparing the maps of AOD overlayed on top of the true color image (Figure 9a,b), the missing MAIAC retrieval can be possibly due to sampling difference, as the two satellites overpassed this region with a temporal offset. Scattered high AOD from VIIRS can be seen, while MAIAC provides less coverage of valid retrievals within this analysis region. Thus, similar to the previous case, the non-collocations tend to be related to the stricter cloud masking of MAIAC algorithm. To evaluate the overall quality of these non-collocated data, we have filtered the VIIRS-AERONET matchups with no valid MAIAC C6.1 data on the same day in September 2020, this yielded only five matchups with a small positive bias of 0.036. As the number of samples is limited, further validation over a longer period is needed. Overall, it can be seen that the cloudy scene creates a marginal situation for aerosol retrieval with high possibility of non-collocated AODs and differences in the AOD distributions between products.

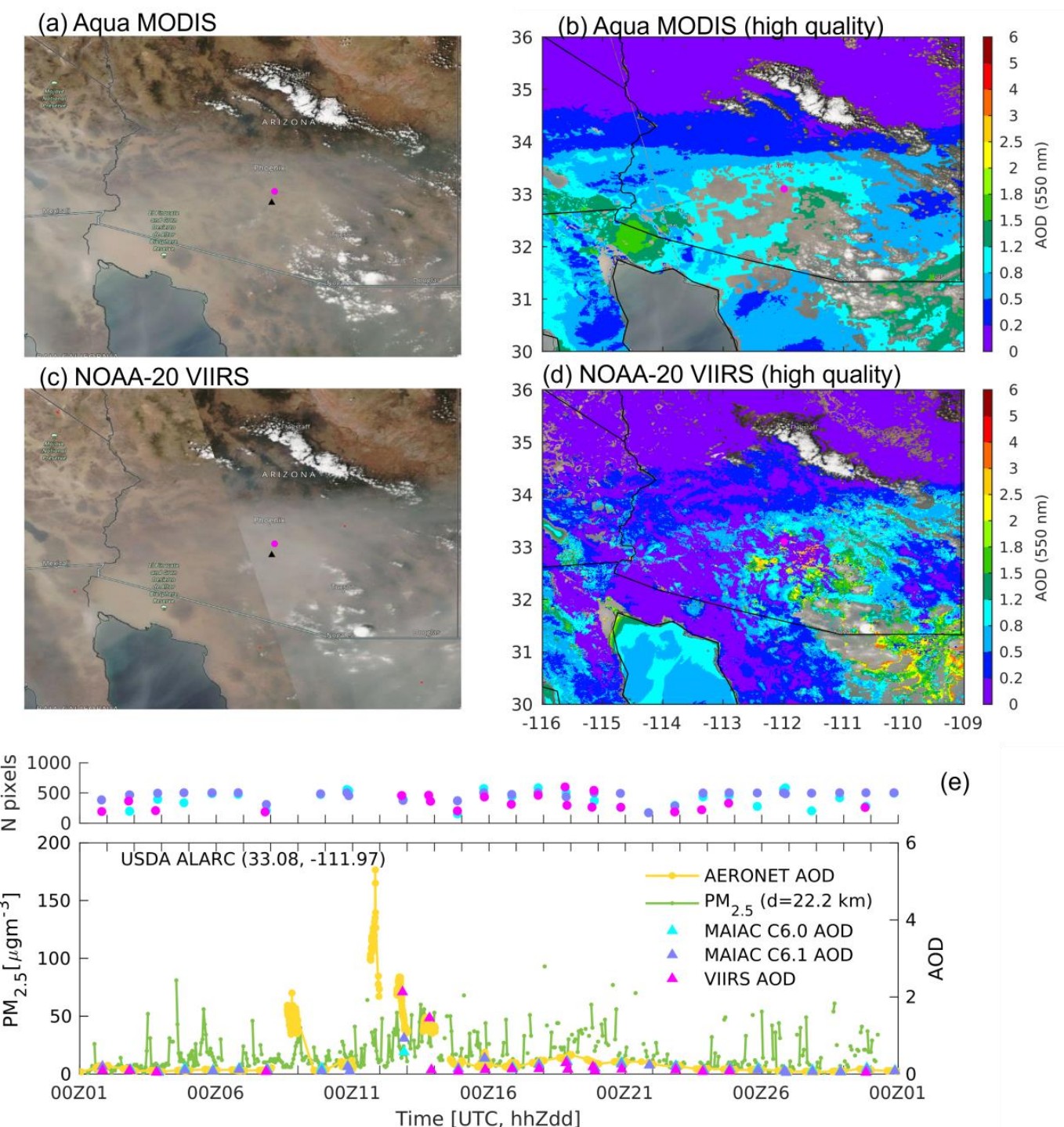

**Figure 8.** Similar to Figure 6, but for 13 September 2020 (**a–e**). The magenta dot stands for location of the AERONET station USDA ALARC. The satellite AODs have been plotted on top of the true color images.

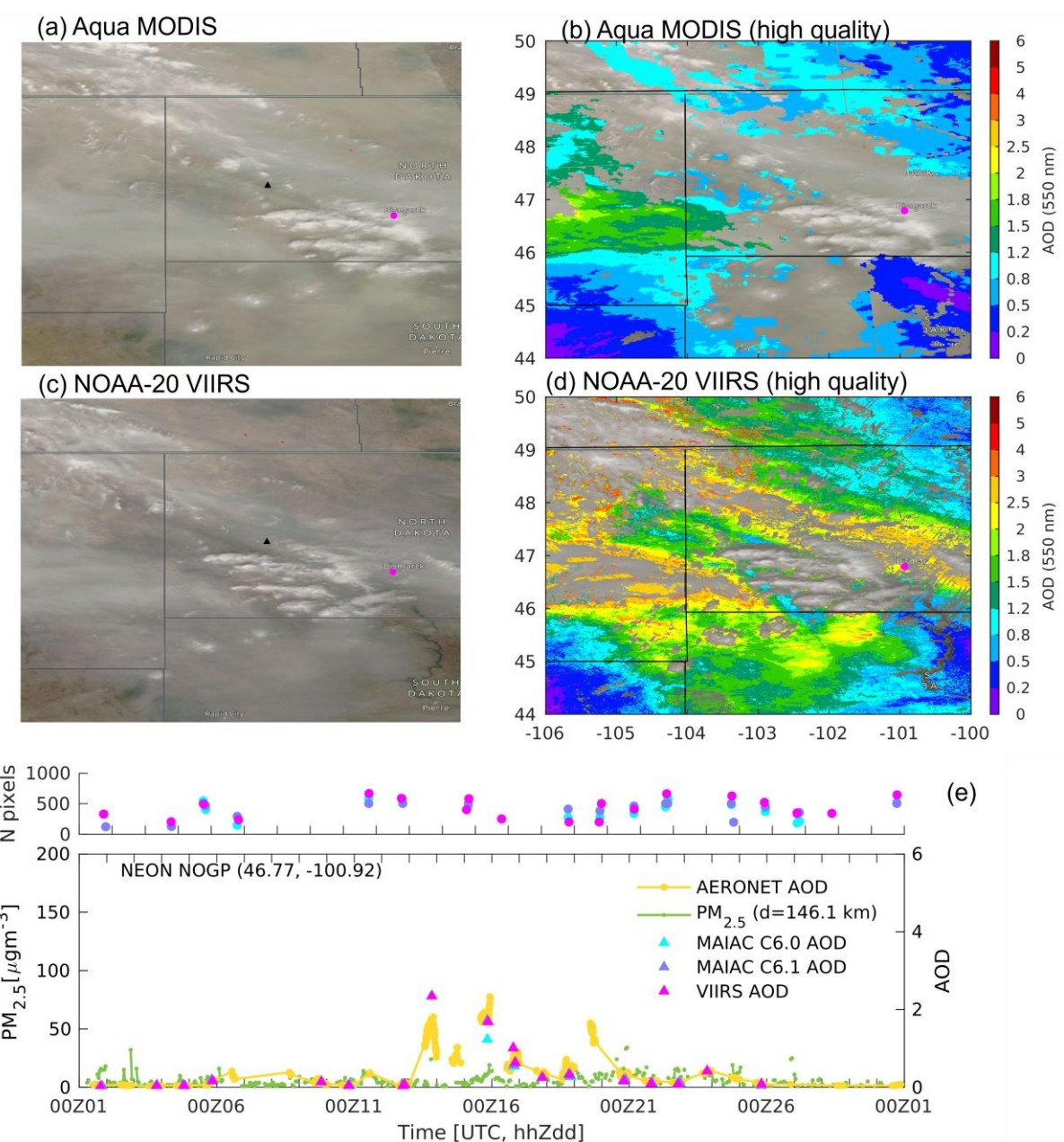

**Figure 9.** Similar to Figure 6 but for 13 September 2020 (**a**–**e**). The magenta dot stands for location of the AERONET station NEON NOGP. The satellite AODs have been plotted on top of the true color images.

## 4. Conclusions

Aiming at validating satellite AOD retrievals with a focus on their accuracy under wildfire smoke conditions, we evaluate and compare the AOD (550 nm) retrievals from the latest version of Terra and Aqua MODIS MAIAC C6.1, its predecessor version MAIAC C6.0, and the NOAA-20 VIIRS EPS algorithm product (v2r3) during September 2020 over the western U.S. We validate the products against AERONET measurements and compare their daily 0.1°-gridded maps. Several cases of retrievals from Aqua MODIS MAIAC C6.1 and VIIRS are analyzed to gain insights on their differences. The main findings are summarized as follows.

Validation of the satellite AOD products is implemented individually for each dataset against the AERONET data at 41 stations. MAIAC C6.1 presents an overall agreement with AERONET with a mean bias of 0.029 and about 73% of retrievals falling within the EE of

$\pm(0.05 \pm 15\%$ AOD). Compared to its preceding version C6.0 that exhibited significant underestimations of AOD for smoke conditions, substantial improvement is achieved for MAIAC C6.1, indicating effectiveness of the algorithm updates on aerosol models. VIIRS data show comparable performance to MAIAC C6.1 with a slightly larger positive bias than its reported global validation. The bias is shown to increase at higher AOD level over the AERONET AOD range of 0.5–3.0. In addition, the spread of retrieval bias tends to be larger with higher AERONET AOD for all algorithms, implying the increased uncertainty of retrievals under high smoke aerosol loading. The validation of collocated retrievals among all three datasets suggests consistent results with the evaluation of each individual product. In addition, the merged data using collocated retrievals from MODIS MAIAC and VIIRS show the potential to provide a better accuracy.

Gridded data at 0.1°-resolution are compared to gain further insights on data coverage and retrieval strategy. On monthly scale, the average AOD from Aqua MODIS MAIAC C6.1 and VIIRS both provide nearly complete spatial coverage over the analysis domain. VIIRS shows fewer retrievals over arid and sparsely vegetated surfaces, in consistency with previous studies that indicated relatively limited data coverage over bright surfaces. This again suggests the method of characterizing surface reflectance as an important contributor to data coverage over arid areas.

Similar monthly AOD distribution patterns are presented among the products, albeit with differences in magnitude. MAIAC C6.1 shows overall higher AODs than C6.0, largely owing to the enlarged retrievals at high aerosol loading over smoke areas. Meanwhile, VIIRS exhibits a higher domain average AOD compared to MAIAC C6.1. The positive differences are most discernable for areas in the northwest and central north, which is the main transport pathway of the smoke plumes over the western U.S during this period of interest. This can be partly explained by the greater retrieval biases towards higher AOD levels. Differences in aerosol models and representations of surface reflectance anisotropy properties are likely the crucial factors driving the discrepancies under these high AOD conditions [29].

Discrepancies in data availability are analyzed between Aqua MODIS MAIAC C6.1 and VIIRS on daily scale, with a specific focus on their availability over smoke areas and transported plumes. Over thick smoke adjacent to the fire sources, MAIAC often provides retrievals, but VIIRS data tends to be identified as low or medium quality. The low or medium quality retrievals are demonstrated to be likely attributed to pixels exceeding the upper limit of 5.0, or not passing internal quality masks, such as cirrus, inhomogeneity, cloud shadow, and cloud adjacency. Nevertheless, cloud masking is not supposed to be a substantial contributor to the fewer high-quality retrievals from VIIRS, given the better accuracy of cloud masking of VIIRS under smoke conditions, as shown by the validation using CALIOP data. Therefore, these results highlight the need of further validating and improving the retrieval quality assessment and internal filters over biomass burning smoke.

Over transported smoke plumes near scattered clouds, non-collocated retrievals are found in cases with VIIRS reporting valid data but MAIAC data being unavailable. The higher spatial resolution and pixel-level processing procedure of VIIRS data could be the contributors, which allow higher chances of sampling clear scenes. In addition, it also tends to be associated with the stricter and more conservative cloud masking of the MAIAC algorithm. This is expected to be improved in a future version of the MAIAC product.

The assessment of accuracy and differences between the satellite AOD products provide useful information on their applications and algorithm improvements. The discrepancies in data coverage would have impacts on the representativity of AOD at a coarser grid with relevance to regional and global models. Heavy smoke and cloudy scenes are found to lead to a higher probability of non-collocations, implying likely contributions by internal masking, sampling differences, geometry characteristics of the instrument, and near-cloud effects on retrieval availability and accuracy. Therefore, the cloud masking and quality assessments need to be further compared and validated to improve the consistency and accuracy under wildfire smoke conditions. This would also help improve the continuity

of MODIS AOD products in the future. In addition, the differences in retrieval biases of MODIS MAIAC and VIIRS EPS AOD products, their dependency on AOD level, and the greater spread on higher AOD are important for data assimilation systems and model evaluation. These need to be taken into consideration when implementing bias correction schemes and characterizing retrieval uncertainties over smoke areas. It would also affect the analysis of climatological AOD trends if different products were combined to provide continuous records. Finally, the critical role of objective representation of smoke aerosol optical properties in the algorithm aerosol models have been highlighted. Further work extending the evaluation to other regions of biomass burning and other fire seasons would provide more comprehensive characterization of performance of the products over a wider range of smoke aerosol source regions and surface properties.

**Author Contributions:** Conceptualization, P.S. and X.Y.; methodology, P.S. and X.Y.; formal analysis, X.Y. and M.D.; data curation, A.L., Y.W. and S.K.; writing—original draft preparation, X.Y.; writing— review and editing, P.S., A.L., Y.W. and S.K.; visualization, X.Y. and M.D.; supervision, P.S.; project administration, P.S., A.L. and S.K.; funding acquisition, P.S., A.L. and S.K. All authors have read and agreed to the published version of the manuscript.

**Funding:** This work has been supported by the following grants: NSF 2013461, NASA 80NSSC18K0629, and NOAA NA18OAR4310107. The authors are grateful to the AERONET PIs for maintaining the sites. The work of A. Lyapustin and Y. Wang was supported by the NASA programs "Science of Terra, Aqua and SNPP" and "SNPP and JPSS Standard Products for Earth System Data Records" (manager M. Falkowski).

**Data Availability Statement:** The MODIS MAIAC C6.0 aerosol product (MCD19A2) is available online (https://lpdaac.usgs.gov/products/mcd19a2v006/, accessed on 30 March 2020). The MODIS MAIAC C6.1 is a research version product and is available from Dr. Alexie Lyapustin and Yujie Wang upon request. The VIIRS AOD product is obtained from the NOAA Comprehensive Large Array-Data Stewardship System (CLASS) (https://www.avl.class.noaa.gov/saa/products/search?datatype_family=JPSS_GRAN, accessed on 30 March 2020). The AERONET aerosol dataset was obtained from https://aeronet.gsfc.nasa.gov, accessed on 13 October 2022. The CALIOP data were obtained from the NASA Langley Research Center Atmospheric Science Data Center (https://www-calipso.larc.nasa.gov/products/, accessed on 20 April 2020). Surface $PM_{2.5}$ observations are available at OpenAQ (https://openaq.org, accessed on 13 October 2022) and U.S. EPA's Air Data (https://www.epa.gov/outdoor-air-quality-data, accessed on 20 April 2020).

**Conflicts of Interest:** The authors declare no conflict of interest. The funders had no role in the design, execution, interpretation, or writing of the study.

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
