# Peer review of "Assessment of Satellite AOD during the 2020 Wildfire Season in the Western U.S."

_remotesensing, doi:10.3390/rs14236113_

Round 1

Reviewer 1 Report

Review of Assessment of satellite AOD during the 2020 wildfire season in the Western US

General comment

This paper presents an evaluation of three satellite AOD products derived from MODIS and VIIRS instruments during the 2020 extreme wildfire event in the Western U.S. It provides scientifically relevant results for AOD user community by investigating the impact of differences in instrument and retrieval algorithm on AOD retrieval accuracy in case of extreme biomass burning event. Particularly, the impact of the accuracy of cloud detection in case of heavy smoke conditions is a key issue for aerosol retrieval and the analysis presented in this paper is a very good contribution. I would suggest to better emphasize this aspect in the text which is the main innovation brought by this paper.

However, the text needs to be improved prior publication. Some statements are not clear enough, some points need to be better emphasized, part of the methodology is missing, the structure of the paper needs improvement. I suggest adding a dedicated section on the methodology and a discussion separated from the result section. The method statements which are currently included in the result section need to be moved to a new method section. The discussion should be consolidated by answering few key issues such as the impact of cloud detection accuracy in case of heavy smoke condition. 

Specific comments

-        Introduction

o   The authors should be more precise when referring to the VIIRS AOD product because there are a NASA and a NOAA product which rely on distinct algorithms.

-        Section 2

o   Check the section number

o   I suggest including a table presenting the main characteristics and differences between AOD products or versions of AOD product.

o   The text should underline the differences between products that can support the discussion

o   Section MODIS MAIAC C6.0: 

§  I suggest focusing on the MAIAC retrieval algorithm characteristics, the general sentences about MODIS coverage or comparison of LEO and GEO orbits are known issues and can be found in the appropriate papers.

§  Page 4:

·       “Given that the surface reflectance …reflectance”: This sentence is too long and it is difficult to understand. Need to be rephrased.

·       Quality filter assessment process for MAIAC is not clearly introduced.

·       What is the main advantage of MAIAC versus DT or DB ? is it a more accurate cloud contamination ? a finer spatial resolution ?

·       The assumptions used in MAIAC should be better emphasized and discussed.

·       Some descriptions are too vague: for example what are the characteristics of aerosol models used in MAIAC (how many ? shape assumption ?, size distribution ?)

·       Regarding the EE equation: Does AOD refer to retrieved AOD or AERONET AOD ? For MODIS DT it refers to AERONET AOD

o   VIIRS AOD

§  The most UpToDate reference is the ATBD (year 2020) that should be quoted here.

§  I strongly suggest checking the description of the algorithm (number and nature of aerosol model over land) given here with the ATBD.

§  Why does the VIIRS dataset include only NOAA20 observations and no SNPP

o   A subsection describing how the retrieval collocation is done is missing

-        Section 3 : results

o   General comments

§  Methodology descriptions should be moved to a dedicated section

§  This section should focus only on the presentation of the results which could be interpreted and discussed in a separate section

o   Specific remarks

§  Page 7:” Therefore, we also examined the quality of VIIRS data by retain-ing all-quality retrievals that passed the internal heavy aerosol test and report AOD > 0.5”: this sentence is not clear , do you mean by “we also examined”. What does it imply for QA filtering of the VIIRS data?

§  Page 7: “the error envelope (EE) of ± (0.05 ± 15 % AOD) from the operational DT product” a reference is needed here, there is a document in the DT MODIS product website that provides the updated EE for C6.1.

§  The normalised metrics should be properly defined. Again, this should be done in a subsection of the new methodology section.

§  Page 7: “which likely indicates the better capability of MAIAC to retrieve over the bright surfaces,.” This interpretation is speculative, there are no evidences that the NOAA VIIRS product more frequently fails over bright land surfaces. 

§  Page 7: “as many of the AERONET sites are located in arid areas (Superczynski et al., 2016)” What is the reason for this ? Is it specific to the study area ? This sampling issue should be acknowledged in the AERONET section and clearly discussed if vegetated dark surface are underrepresented compared to bright surfaces.

§  Page 7, section 3.1.1: the first paragraph describing the collocation with AERONET data should be moved to the new method section

§  Page 7: Not sure that Laszlo and Liu, 2017 validation study is the most recent one, this paper investigated an older version of the VIIRS NOAA dataset.

§  Page 8-9, section 3.1.2: this section it is not clear and need to be rewritten: reading the first paragraph I understand that the comparison against AERONET is done by taking the collocated satellite AOD. In the second paragraph, a merged VIIRS-MAIAC 6.1 product is evaluated which is a different perspective. 

§  Page 10, 3.2.1: The choice of reprojecting the products on a regular grid at coarser spatial resolution than their native resolution and the use of monthly average to compare them needs further justification. This is equivalent to a level-3 comparison which may provide different statistics than level-2 intercomparison. This has to be acknowledged and discussed properly using references to recently published AOD product intercomparison studies.

§  Page 10, Section 3.2.1: the first paragraph of collocation should be move to a method section

§  Page 10: “Meanwhile, the number of retrievals from Terra MODIS are overall larger than those for Aqua MODIS, which is likely related to the more frequency of cloudy conditions in the afternoon”: This is valid over land but may be season dependent.

§  Page 12, first paragraph: Another difference between MODIS and VIIRS geometry is the smaller deformation of pixel size along the swath for VIIRS compared to MODIS

§  Page 13-13: 3..2.2 Cloud masking: This analysis is very relevant and should be better emphasized as it is the main outcome of this study.

-        Conclusion:

o   There is no need to justify the study (first sentence), this belongs to Introduction

-        Figures

o   Overall good quality

o   Figure 6 and 8: what is the benefit of showing PM2.5 ? it disturbs the visual comparison of AOD data 

Author Response

Thank you! 
